# Mindfulness-Based Interventions for Physical and Psychological Wellbeing in Cardiovascular Diseases: A Systematic Review and Meta-Analysis

**DOI:** 10.3390/brainsci11060727

**Published:** 2021-05-29

**Authors:** Flavia Marino, Chiara Failla, Cristina Carrozza, Maria Ciminata, Paola Chilà, Roberta Minutoli, Sara Genovese, Alfio Puglisi, Antonino A. Arnao, Gennaro Tartarisco, Flavio Corpina, Sebastiano Gangemi, Liliana Ruta, Antonio Cerasa, David Vagni, Giovanni Pioggia

**Affiliations:** 1Institute for Biomedical Research and Innovation (IRIB), National Research Council of Italy (CNR), 98164 Messina, Italy; flavia.marino@cnr.it (F.M.); chiara.failla@irib.cnr.it (C.F.); cristina.carrozza@irib.cnr.it (C.C.); paola.chila@irib.cnr.it (P.C.); roberta.minutoli@irib.cnr.it (R.M.); sara.genovese@cnr.it (S.G.); alfio.puglisi@irib.cnr.it (A.P.); antoninoandrea.arnao@cnr.it (A.A.A.); gennaro.tartarisco@cnr.it (G.T.); fcorpina@cotmessina.it (F.C.); liliana.ruta@cnr.it (L.R.); david.vagni@irib.cnr.it (D.V.); 2Classical Linguistic Studies and Education Department, Kore University of Enna, 94100 Enna, Italy; 3Departamento de Estudios Románicos, Franceses, Italianos y Traducción (ERFITEI), Universidad Complutense Madrid, 28040 Madrid, Spain; mciminat@ucm.es; 4Multi-Specialist Clinical Institute for Orthopaedic Trauma Care (COT), 98124 Messina, Italy; 5Operative Unit of Allergy and Clinical Immunology, Department of Clinical and Experimental Medicine, School of Allergy and Clinical Immunology, University of Messina, 98125 Messina, Italy; gangemis@unime.it; 6Institute for Biomedical Research and Innovation (IRIB), National Research Council of Italy (CNR), 87050 Mangone, Italy; antonio.cerasa@cnr.it; 7S’Anna Institute, Research in Advanced Neurorehabilitation (RAN), 88900 Crotone, Italy

**Keywords:** mindfulness, cardiovascular diseases, psychological outcome, physical outcome, systematic review

## Abstract

Background: Recently, there has been an increased interest in the efficacy of mindfulness-based interventions (MBI) for people with cardiovascular diseases (CVD), although the exact beneficial effects remain unclear. Methods: This review aims to establish the role of MBI in the management of wellbeing for patients with CVD. Seventeen articles have been included in this systematic synthesis of the literature and eleven in the meta-analysis. Results: Considering physical (i.e., heart rate, blood pressure) and psychological outcomes (i.e., depression, anxiety, stress, styles of coping), the vast majority of studies confirmed that MBI has a positive influence on coping with psychological risk factors, also improving physiological fitness. Random-effects meta-analysis models suggested a moderate-to-large effect size in reducing anxiety, depression, stress, and systolic blood pressure. Conclusions: Although a high heterogeneity was observed in the methodological approaches, scientific literature confirmed that MBI can now be translated into a first-line intervention tool for improving physical and psychological wellbeing in CVD patients.

## 1. Introduction

Cardiovascular disease (CVD) is among the leading causes of disability and mortality globally [1]. It is well known that this disorder is associated with a particular lifestyle, such as smoking, incorrect feeding, hyperlipidemia, hyperglycemia, hypertension, obesity, alcohol intake, and diabetes [2]. Furthermore, psychosocial stress is widely recognized as one of the main contributors to all these risk factors; therefore, behavioral approaches to primary and secondary prevention are gaining research support. According to the World Health Organization (WHO) [3], actions aimed at reducing these major risk factors can drastically impact disability and death from CVD.

Mindfulness-based interventions (MBI) have been proposed as an efficient and easily applicable additional complementary treatment for reducing psychological stress in the management of various physical and mental health conditions [4,5]. MBI generally refers to practices that cultivate awareness and require paying attention to the present moment [6]. Mindfulness training strengthens metacognitive awareness (self-reflective capacity to monitor mental experience), allowing participants to shift their perspective (“reperceiving”) and reduce emotional reactivity [7]. The two main practices of MBI are Mindfulness-Based Stress Reduction (MBSR) and Mindfulness-Based Cognitive Therapy (MBCT). MBSR has been used since 1979 as a training vehicle for the relief of pain and distress in people with chronic health problems [8]. In general, this program includes 8 weekly meetings of 1–2 h approximately and 8 final hours of a full-day retreat [9,10]. MBSR has been found to have positive effects on pain, anxiety, and stress in people with chronic disorders such as fibromyalgia, cancer, arthritis, and coronary artery disease [11,12,13]. On the other hand, MBCT was initially developed as a relapse prevention treatment for those with a high risk of recurring depression, but it has further been adapted to a range of different clinical realms. This protocol teaches the subject to move away from ruminative mental patterns typical of depression and to adopt a new perspective in which thoughts and feelings are considered transitory rather than objective representations of reality [14].

To manage physical and psychological symptoms in CVD patients, meditative practices have been proposed, in addition to conventional medical interventions, as an effective approach for reducing clinical complications in CVD patients [15,16]. However, some inconsistent findings have been reported. Indeed, some studies have found that patients with CVD appear to benefit only psychologically from mindfulness interventions [15], whereas others suggest possible benefits also for physical risk factors, such as blood pressure, obesity, and smoking [2]. Currently, few studies have shown combined physical and psychological positive effects [17], while the vast majority described mixed results [15].

For this reason, we sought to perform a new synthesis of literature aimed at re-examining the question of whether there is sufficient evidence to conclude that MBI can be included in clinical recommendations for primary and secondary prevention of CVD.

## 2. Materials and Methods

This review was planned and conducted in accordance with Preferred Reporting Items for Systematic Reviews and Meta-Analyses (PRISMA) guidelines [18]. Articles published between 2003 and 2019 were reported using electronic bibliographic databases such as PubMed, Science Direct, and Google Scholar. To improve the search strategy, keywords including “text words” and MeSH were used. The search terms incorporated the following keywords: “mindfulness, meditative practices, cardiovascular diseases, heart failure, congestive heart failure, coronary heart disease, cardiovascular abnormalities, congenital abnormalities, chronic heart failure, a heart defect and cardiovascular infection” (Table 1). The research included only original research in the English language and peer reviews. Unpublished dissertations, book chapters, and conference papers were excluded.

Criteria for including or excluding papers were determined a priori. Papers were considered for inclusion only if (a) they were written in full-text English language in a peer-reviewed journal; (b) they were published between 2003 and 31 December 2019; and (c) they included subjects with cardiovascular diseases, coronary heart disease, congestive heart failure, congenital heart diseases, and chronic heart failure. Studies in which the sample presented with risk factors for the development of cardiovascular diseases (for example, pre-hypertension, hypertension, obesity, smoke, etc.) were also considered. Articles were excluded if (a) they considered subjects with a history of other neurological or psychiatric disorders; or (b) they considered other meditative practices such as transcendental meditation, autogenic training, or yoga.

The data collected from each article were categorized as information on the first author and year of publication, the size of cohorts, the modalities of intervention, the type of clinical disorders, the study design, the outcomes (physical and psychological), the main results, and conclusions.

Following the Cochrane guidelines, the Quality Assessment of Diagnostic Accuracy Studies (QUADAS) tool [19] was used to assess the methodological quality and the risk of bias of each study. This quality assessment allowed classifying studies as having a low, high, or unknown risk of bias.

### Meta-Analyses

Additional inclusion criteria were considered for the meta-analyses: (1) Participants should not overlap among studies; (2) studies reporting an explicit intervention protocol should be tested against a control group; (3) studies should study a population with comparable ages with the ones reported in at least 2 other studies.

Inclusion criteria used to select measures and constructs to analyze among the ones reported by the included studies were: (1) Outcome measures should have been taken <1 month after the study ended; (2) a construct or related measures should have been explored by at least 3 studies; (3) in studies with more than 2 groups, only MBI and control groups were selected.

Separate meta-analyses were conducted for each construct; the uncorrected *p*-value was set at 0.05 but was to be lowered in accordance with Sidak correction for multiple discoveries depending on the number of constructs that were included in the study in the end. Hedges’ *g* was used as an estimate of effect size.

Outcome measures were homogenized among studies so that positive *g*-values are always positive outcomes.

Even if we used additional criteria for the meta-analyses, we expected heterogeneity in treatment effects caused by differences in study populations (such as the age of patients), interventions received (such as online or in person), length of treatment, and other factors. Therefore, we chose a random-effects meta-analysis model because we assumed that the observed estimates of treatment effect can vary across studies due to the real differences in the treatment effect in each study and not only to sampling variability (chance) as assumed by the fixed model. We ran the fixed model anyway and report the data for completeness. Hedges’ *g* was used as an estimate of effect size. The *I* square (*I*^2^) statistic represents the percentage of variability in effect estimates that is due to heterogeneity. The *I*^2^ index can be interpreted as the percentage of total variability in a set of effect sizes due to true heterogeneity (between-studies variability). [20] In random-effects meta-analysis, the extent of variation among the effects observed in different studies (between-study variance) is referred to as tau-squared (τ^2^) [20]. τ^2^ is the variance of the effect size parameters across the population of studies, and it reflects the variance of the true effect sizes.

## 3. Results

### 3.1. Study Selection

The electronic bibliographic database search strategy of five databases retrieved 5600 studies. We screened titles/abstracts and adjusted for duplicates, and N = 115 studies remained after the reviewing process. From this group, 61 studies were further excluded because they focused on other physical and/or psychological conditions. In the second phase, 54 studies were excluded because they did not fulfill inclusion criteria. Indeed, 24 studies used other meditative practices (such as yoga and transcendental meditation), and 13 studies were excluded for comorbidity with other physical and psychological conditions. Finally, 17 articles were included in this review (Figure 1).

### 3.2. Study and Sample Characteristics

The trial size ranged between 45 and 382 participants, with 5 out of 24 trials having >100 participants. The mean age of participants recruited ranged between 12 and 76 years. In total, 1611 participants were enrolled across 17 trials, and 16 out of 17 studies reported the gender of participants (women vs. men). Among the studies, 13 recruited both male and female participants, while 2 reported a sample of only female participants and 1 study of only males. All studies (N = 17) examined a control group (waitlist or a different mindfulness intervention) (Table 2).

Six studies are reported in the review but were removed in the meta-analysis due to exclusion criteria. One study with teenage participants (mean age = 14.5) [21] was excluded because the participant age was too different than the others. Another was a 12-month follow-up [22] of an already included study [17], 2 studies were a single section mindfulness experimental study to measure instant hearth measure changes [23,24], and 2 studies used outcome measures taken by less than 2 other studies [25]. The 11 studies selected for meta-analysis reported 18 different constructs, but only 6 were reported in 3 or more studies. Selected constructs were anxiety, depression, perceived stress, quality of life, and measured blood pressure (diastolic and systolic). Each outcome was measured by 4 to 7 studies. The *p*-value was set to 0.009 (Table 3).

### 3.3. Mindfulness and Physical Outcomes

All studies confirmed the effectiveness of MBI-related treatments on the physical outcomes of CVD patients.

Five studies measured blood pressure. There was a large reduction in systolic blood pressure (*g* = 1.12, 95% CI: 0.40; 1.84, *p* = 0.002, *I^2^* = 89.1%, τ^2^ = 0.56), while diastolic blood pressure was not significant after correction (*g* = 0.66, 95% CI: −0.04; 1.36, *p* = 0.043, *I^2^* = 89.3%, τ^2^ = 0.54) [9,17,26,27,28] (Table 4, Figure 2).

Furthermore, one study also reported a significant reduction in heart palpitations (F = 14.4, *p* < 0.001) and heart rate (effect size, beats per minute: −2.8, 95% CI: −5.4; −0.2, *p* = 0.033). Significant improvements were further detected in physical risk factors such as fatigue (effect-size using FSS sum-change on the group = –8.0; *p* = 0.0165), unsteadiness/dizziness (*p* = 0.039), breathlessness, and/or tiredness (*p* = 0.0087) [29].

With regard to online mindfulness training, few studies have investigated the effects of this methodology on physical parameters [17,21,22]. Two studies showed a small positive effect on exercise capacity (*d* = 0.22; 95% CI 0.05; 0.39; *p* = 0.055) and heart rate (beats per minute: −2.8, 95% CI: −5.4; −0.2, *p* = 0.033) [17,22].

**Table 2 brainsci-11-00727-t002:** Characteristics of studies assessing the effects of mindfulness on patients with cardiovascular disease.

Reference	Sample	Country	Intervention	Cardiological Disorders	Physiological Measures	Psychological Measures	Main Findings
Ahmadpanah et al. (2016) [9]	MDM (*n* 15 female; Age: 46.3 year) SMT (*n* 15 female; Age: 46.5 year) Control Group (*n* 15 female; Age: 46.5 year)	IRAN	MDM group: 8 weekly sessions; lasting 1 h STM group: 8 weekly sessions, lasting 1 h Control Group: brief medical check once a week for 8 weeks	HT	BP	BDI BAI	MDM and SMT treatments produced improvements in symptoms of anxiety and depression and reduced BP
Doherty et al. (2015) [10]	Treatment Group: *n* 32 (12 women and 19 men; mean age: 57.6 year) Control Group: 30 (8 women and 22 men mean age: 59.6 year)	IRELAND	MBCT group: 8 weekly sessions; each lasted 2 h Control Group: waiting list. Received no psychological intervention until Time 3, when they were offered MBCT. All participants were assessed at baseline (Time 1), 8 weeks later (Time 2), and at 6-month follow-up (Time 3)	CHT		HADS BSI POMS PAIS MAAS PQ CSQ	MCBT showed significant improvement rates in clinical depression.
Freedenberg et al. (2017) * (adolescents) [21]	Treatment group: 26 (18 women and 8 men; mean age: 15.1 year) Video Online Group: 20 (11 women and 9 men; mean Age: 14.5 year)	USA	MBSR group: 6 weekly sessions each lasted an hour and a half Control Group: video online support group met for six consecutive weekly one-hour session	Congenital Cardiac Diagnoses Cardiac device or postural orthostatic tachycardia syndrome		HADS RSQ	MBSR intervention did not induce significant improvements in anxiety and depression scores
Gotink et al. (2017) * (follow-up) [22]	MBSR group: (*n* 215; 44.2% female and 55.8% male; mean age: 43.2 year) Control Group: (*n* 109; 50.5% female and 49.5% male; mean Age: 43.2 year)	JAPAN	MBSR group: 12 weekly sessions online Control Group: usual care by their treating cardiologist	Heart disease	BP Heart rate Respiratory rate Cortisol level	SFHS Visual Analogue Scale Hospital Anxiety and Depression scale Perceived Stress Scale Perceived Social Support Scale	Exercise capacity, systolic blood pressure, mental functioning, and depression improved significantly compared to UC
Grant et al. (2013) * (not an intervention protocol) [23]	MBI group: 48 (50% female) Control Group: 49 (57.1% female and 42.9% male)	ALBANY	MA: focused-breathing exercise Control Group: no breathing exercise	Hypertension	Systolic and Diastolic BP Heart rate		There were no group differences in reactivity to either stressor. Participants in the mindfulness-analog condition experienced significantly greater latency to systolic blood pressure recovery following the CPT and a tendency toward greater latency to diastolic blood pressure recovery
Hughes et al. (2013) [26]	MBSR Group (*n* 28; 61% women; mean age: 51.2 year Control Group: (*n* 28; 54% women; mean age: 49.5 year)	UK	MBSR Group: 8 weekly sessions; each lasted 2.5 h Control Group: progressive muscle relaxation (PMR) 8 weekly sessions; each lasted 2.5 h	Prehypertension	Systolic and Diastolic BP		MBSR is effective in lowering Systolic and Diastolic BPe in prehypertensive individuals.
Jalali et al. (2019) [30]	MBSR group: (*n* 30; 50% female) Control group (*n* 30, 50% female)	IRAN	MBSR group: 8 weekly sessions each lasted 2 h Control Group: routine medical care	Cardiovascular disease		Self-Efficacy (General Self-Efficacy Scale) Quality of life (Short Form Survey)	MBSR program led to a stable improvement of the scores of self-efficacy and quality of life in the experimental group
Momeni et al. (2016) [28]	MBRS group (*n* 30 13 female and 17 male Age: 49.2 year) Control Group (*n* 30 12 female and 18 male; Age: 46.2 year	IRAN	MBSR group: 8 weekly sessions lasting 2.5 h Control Group: receive no psychological therapy, waiting list	Cardiovascular disease	BP AOBP	PSS-14 STAI-X	MBSR helps people to deal with stress, pain, and illness more effectively and play a more active role in their lives and recovery. MBSR was also effective in reducing cardiac patients’ BP, perceived stress, and anger.
Nehra et al. (2014) [13]	MBRS group (*n* 25) Control Group (*n* 25)	INDIA	MBSR group: 8 weekly sessions lasting 2 h Control Group: Standard treatment	Coronary heart disease		PSS HCS	MBSR is highly effective in reducing perceived stress and health complaints in CHD patients.
Norman et al. (2018) [29] * (not yet replicated measures)	MBI group: *n* 22 (11 female and 11 male; Mean Age: 76.5 year) Control Group: *n* 18 (6 female and 12 male; Mean Age: 75.0 year)	Sweden	MBI group: 8-weekly educational and training sessions lasting 2 h Control group: received usual care comprising standard health care for patients with CHF	CHF	Heart rate Respiratory rate Fatigue severity scale Karolinska Sleep Questionnaire Unsteadiness/dizziness		MBI is effective in reducing the self-reported impact of fatigue on daily living, unsteadiness/dizziness, and breathlessness/tiredness related to physical function
Nyklıcek et al. (2014) [31]	MBRS group *n* 55 (10 female and 45 male; Mean Age: 55.4 year) Mindfulness Self-Help Control Group *n* 52 (9 female and 43 male; Mean Age: 56.3 year)	USA	MBSR: group: 3 weekly meetings (lasting 90–120 min) with an additional evaluation session 2 weeks later). Mindfulness Self-Help Control Group: Participants receiving the self-help booklet were asked to thoroughly read the theory and to practice exercises daily	PCI		SAD-4 PSS GMS The World Health Organization Quality of Life-Brief Questionnaire SAQ FMI-s	The group mindfulness intervention reduced perceived stress and symptoms of anxiety and depression more strongly than the control group in the relatively younger subsample.
Owens et al. (2016) [25] * (not yet replicated measures)	17 female and 3 male; Mean Age: 49.4 year	USA	MBSR group: 8 weekly sessions lasting 2.5 h Control Group: Waiting list control	Palpitations	Heart palpitations		MBSR participants reported a significant reduction in heart palpitations, and this improvement in the MBSR participants was sustained at 1-month follow-up.
Parswani et al. (2013) [27]	MBSR group (*n* 15; male mean age = 47.3 year) TAU group (*n* 15 male mean age = 50.6 year)	INDIA	MBS group: eight weekly sessions, lasting 1–1.5 h Control Group: health education session	CHD	BP BMI	Hospital Anxiety and Depression Scale PSS	MBSR treatment produces significant reduction in symptoms of anxiety and depression, perceived stress, BP, and BMI in CHD patients
Steffen et al. (2015) [24] * (not an intervention protocol)	MBI group: *n* 30 (47% female and 53% male Age: 19.9 year) Control Group: *n* 32 (53% female and 47% male Age: 20.6 year)	USA	MBI group: Brief 1-day passive listening of two tracks from CD + Mindfulness of Breathing exercise Control Group: Brief 1-day passive listening of two tracks from CD	Cardiovascular reactivity	BP Heart rate	BDI STAI PASAT Working memory and auditory attention	Mindfulness participants showed lower systolic blood pressure and decreased systolic during cognitively stressing activity, whereas no significant effects were detected for mood levels.
Sullivan et al. [32] (2009)	MBSR group: *n* 108 (66.7% male and 33.3% female Age: 61.5 year) Control Group: *n* 100 (73% male and 27% female Age: 61.1 year)	UK	MBSR group: once a week for 2.25 h for 8 consecutive weeks. Control Group: Usual care treatment	CHF	KCCQ 23-item	CES-D 10-item POMS	MBSR significantly reduced depression and anxiety, while improving overall quality of life and clinical symptoms in patients with CHF when compared to control patients.
Tacòn et al. (2003) [33]	MBSR group: *n* 10 (female Age: 60.5 year Control Group: *n* 10 female Age: 60.5 year	USA	MBSR group: 8 weeks-2 h each week Control Group: Participants were placed on a waiting list and offered the opportunity to participate in the program after the study was completed.	Angina, hypertension, cardiovascular disease, and cardiac valve disorders.		STAI Courtauld Emotional Control Scale Problem-Focused Styles of Coping Multidimensional Health Locus of Control scale	Women in the intervention group showed improvement in anxiety scores and decrease in the control of negative emotions.
Younge et al. (2015) [17]	MBI group: *n* 215 (44.2% female and 55.8% male Age:43.2 year) Control Group: *n* 109 (50.5% female and 49.5 male Age: 43.2 year)	UK	MBI group: 12-week session lasting 1 h Control group: usual care by their treating cardiologist. Treatment and frequency differ between patients	Heart disease	Heart rate BP Respiratory rate NT-proBNP	The Dutch version of the Perceived Stress Scale Social Support Scale 12 Blumenthal	Online mindfulness training shows positive effects on heart rate, systolic and diastolic blood pressure. Whereas no significant effect was found on psychological outcomes.

* Excluded from the meta-analysis. MBCT: Mindfulness-Based Cognitive Therapy; MBSR: Mindfulness-Based Stress Reduction; MBI: Mindfulness-Based Intervention; metacognitive detached mindfulness therapy (MDM) and stress management training (SMT); MA: Mindfulness Analog; BDI: Beck Depression Inventory; BAI: Beck Anxiety Inventory; HADS: Hospital Anxiety and Depression Scale; BSI: Brief Symptom Inventory; POMS: Profile of Mood States; PAIS: Psychosocial Adjustment to Illness Scale; MAAS: Mindful Attention Awareness Scale; PQ: Questionnaire on helpful aspects of therapy; CSQ: Client Satisfaction Questionnaire; HADS: Hospital Anxiety and Depression Scale; RSQ: Responses to Stress Questionnaire; SFHS: Short-Form Health survey; PSS-14: Cohen’s Perceived Stress Scale; PSS: Perceived Stress Scale; HCS: Health Complaints Scale; SAD-4: The Symptoms of Anxiety–Depression index; GMS: Dutch Global Mood Scale; SAQ: The Seattle Angina Questionnaire; FMI-s: Freiburg Mindfulness Inventory; STAI: State-Trait Anxiety Inventory; PASAT: Paced Auditory Serial Addition Task; CES-D 10: Center of Epidemiology—Depression; POMS: Profile of Mood States; BP: blood pressure; AOBP: Automated Office BP measurement; KCCQ: Kansas City Cardiomyopathy Questionnaire; CHF: chronic heart failure; PCI: percutaneous coronary intervention; CHD: coronary heart disease; HT: hypertension; CHT: chronic heart disease.

**Table 3 brainsci-11-00727-t003:** Statistical results for studies included in the meta-analysis.

Construct	Study	MBI Group			Control Group			Cohen’s *d*	Hedges’ *g*				Weight (%) Fixed Model	Weight (%) Random Model
		N	MD	SD	N	MD	SD		*g*	SE	95% CI			
Anxiety	Tacòn et al. (2003) [33]	10	−8.77	9.31	10	0.330	12.8	0.814	0.779	0.446	−0.094	1.65	3.26	14.1
	Sullivan et al. (2009) [32]	108	−1.25	8.16	100	1.80	4.14	0.466	0.464	0.140	0.190	0.739	33.0	19.8
	Parswani et al. (2013) [27]	15	−1.74	3.30	15	−0.140	3.49	0.471	0.458	0.360	−0.248	1.16	5.00	15.8
	Doherty et al. (2015) [10]	32	−5.37	4.07	30	−1.63	4.30	0.894	0.883	0.263	0.367	1.40	9.36	17.8
	Younge et al. (2015) [17]	215	−0.500	3.20	109	−0.900	3.00	−0.128	−0.127	0.117	−0.357	0.103	47.0	20.1
	Ahmadpanah et al. (2016) [9]	15	−6.20	4.57	15	7.87	4.54	3.09	3.01	0.526	1.98	4.04	2.34	12.5
Depression	Sullivan et al. (2009) [32]	108	−1.74	17.3	100	0.46	14.24	0.139	0.138	0.138	−0.133	0.409	32.0	17.8
	Parswani et al. (2013) [27]	15	−2.80	1.82	15	0.54	2.44	1.55	1.51	0.405	0.714	2.30	3.74	15.5
	Nyklicek et al. (2014) [31]	55	−1.61	0.450	52	−0.210	.460	3.08	3.06	0.284	2.51	3.62	7.61	16.7
	Doherty et al. (2015) [10]	32	−5.49	3.67	30	−1.73	4.36	0.936	0.924	0.264	0.406	1.44	8.79	16.9
	Younge et al. (2015) [17]	215	−0.50	2.90	109	0.000	2.30	0.184	0.184	0.118	−0.047	0.414	44.5	17.7
	Ahmadpanah et al. (2016) [9]	15	−8.00	4.52	15	1.33	5.22	1.91	1.86	0.429	1.02	2.70	3.34	15.2
Perceived Stress	Parswani et al. (2013) [27]	15	−10.5	4.13	15	−2.74	8.58	1.16	1.13	0.384	0.373	1.88	5.68	18.6
	Nehra et al. (2014) [13]	25	−3.16	2.62	25	−0.920	3.60	0.712	0.701	0.287	0.138	1.26	10.1	19.9
	Nyklicek et al. (2014) [31]	55	−4.5	1.04	52	−2.05	1.05	2.30	2.29	0.248	1.80	2.77	13.7	20.3
	Younge et al. (2015) [17]	215	−2.40	6.30	109	−0.900	6.80	0.232	0.231	0.118	0.001	0.462	6.47	21.4
	Momeni et al. (2016) [28]	30	−13.5	7.57	30	−4.04	4.60	1.51	1.49	0.289	0.926	2.06	10.0	19.8
Quality of Life	Nyklicek et al. (2014) [31]	55	−2.55	0.370	52	−0.630	.410	4.92	4.88	0.385	4.13	5.64	6.37	24.4
	Doherty et al. (2015) [10]	32	−10.3	16.15	30	1.90	18.9	0.693	0.685	0.258	0.178	1.191	14.2	25.1
	Younge et al. (2015) [17]	215	0.40	10.40	109	0.700	9.30	0.030	0.030	0.117	−0.200	0.260	68.6	25.5
	Jalali et al. (2019) [30]	30	−13.0	8.35	30	0.970	8.59	1.65	1.63	0.295	1.05	2.21	10.8	24.9
Blood Pressure	Hughes et al. (2013) [26]	28	−2.40	5.30	28	1.10	7.11	0.558	0.550	0.269	0.024	1.08	12.3	20.9
(Diastolic)	Parswani et al. (2013) [27]	15	−2.56	5.34	15	−1.60	5.50	0.177	0.172	0.356	−0.525	0.870	6.98	19.2
	Younge et al. (2015) [17]	215	−2.34	8.90	109	−3.39	10.1	−0.113	−0.112	0.117	−0.342	0.118	64.2	23.1
	Ahmadpanah et al. (2016) [9]	15	−21.5	10.7	15	9.69	9.17	3.14	3.051	0.530	2.01	4.09	3.14	15.6
	Momeni et al. (2016) [28]	30	−1.66	8.39	30	0.50	6.65	0.285	0.282	0.256	−0.220	0.784	13.48	21.15
Blood Pressure	Hughes et al. (2013) [26]	28	−4.90	6.87	28	−0.70	7.83	0.570	0.562	0.269	0.035	1.089	12.68	20.98
(Systolic)	Parswani et al. (2013) [27]	15	−11.20	11.40	15	10.14	23.77	1.145	1.114	0.383	0.363	1.865	6.24	18.78
	Younge et al. (2015) [17]	215	−5.17	14.50	109	−1.50	15.50	0.247	0.247	0.118	0.016	0.477	66.14	23.10
	Ahmadpanah et al. (2016) [9]	15	−36.33	15.59	15	9.53	17.46	2.771	2.696	0.497	1.721	3.671	3.70	16.46
	Momeni et al. (2016) [28]	30	−15.83	7.73	30	−2.83	10.25	1.432	1.413	0.286	0.854	1.973	11.23	20.68

MBI: Mindfulness-Based Intervention; N: number of participants; MD: mean difference between post-intervention and pre-intervention; SD: standard deviation; CI: confidence Interval.

**Table 4 brainsci-11-00727-t004:** Summary of meta-analysis results—fixed and random-effects models.

Construct	N Studies	Model	Hedges’ g						Heterogeneity			
			*g*	SE	95% CI		*z*-Score	*p*-Value	*I* ^2^	χ^2^	τ^2^	*df*
Anxiety	6	Fixed	0.295	0.081	0.137	0.453	3.66	<0.001 *	0.894	47.34	-	5
		Random	0.780	0.294	0.204	1.38	2.65	0.008 *	0.890	-	0.42	-
Depression	6	Fixed	0.559	0.078	0.405	0.713	7.13	<0.001 *	0.956	113.6	-	5
		Random	1.24	0.421	0.419	2.07	2.96	0.003	0.966	-	0.98	-
Perceived Stress	5	Fixed	0.737	0.091	0.557	0.916	8.05	<0.001 *	0.939	65.47	-	4
		Random	1.16	0.435	0.306	2.01	2.66	0.007 *	0.939	-	0.87	-
Quality of Life	4	Fixed	0.605	0.097	0.415	0.796	6.23	<0.001 *	0.981	159.6	-	3
		Random	1.78	0.878	0.060	3.500	2.03	0.043	0.981	-	3.0	-
Blood Pressure	5	Fixed	0.141	0.094	−0.043	0.325	1.50	0.133	0.893	37.38	.	4
(Diastolic)		Random	0.657	0.357	−0.043	1.36	1.84	0.066	0.893	-	0.54	-
Blood Pressure	5	Fixed	0.563	0.096	0.375	0.750	5.88	<0.001 *	0.891	36.54	-	4
(Systolic)		Random	1.12	0.365	0.404	1.84	3.07	0.002 *	0.891	-	0.56	-

* *p* < 0.009, significant after correction.

### 3.4. Mindfulness and Psychological Outcomes

MBI was effective in reducing mood symptomatology and perception of stress across a variety of clinical populations [9,10,13,22,24,27,28,30,31,33]. Only one study did not describe any significant psychological benefit before and after MBI intervention [17].

The main psychological benefit concerns depression symptomatology. There was a very large effect of MBI on depression among 6 studies included in the meta-analysis (*g* = 1.24, 95% CI: 0.42; 2.07, *p* = 0.003, *I*^2^ = 95.6%, τ^2^ = 0.98) [9,10,17,27,31,32] (Table 4, Figure 2).

O’ Doherty et al. [10] demonstrated that depressed patients with CVD who completed an 8-session MBCT training had an effect size for depressive symptoms of *d* = 0.77 after treatment and *d* = 0.60 at the 6-month follow-up.

Five studies used perceived stress as an outcome measure. In such studies, a significantly large effect of mindfulness in CVD was observed for perceived stress (*g* = 1.16, 95% CI: 0.31; 2.01, *p* = 0.007, *I^2^* = 93.9%, τ^2^ = 0.87) [13,17,27,28,31]. Six studies used anxiety as an outcome variable. In such studies, a significantly moderate effect of mindfulness in CVD was also observed (*g* = 0.78, 95% CI: 0.20; 1.36, *p* = 0.008, *I^2^* = 89.4%, τ^2^ = 0.42) [9,10,17,27,32,33] (Table 4, Figure 2). This effect was also observed in adolescents (*t*(1,9) = 3.67, *p* < 0.01) [34] and partially in young adults (*t*(105) = 1.84, *p* = 0.07) [31]. Similarly, in older adults, there was a significant reduction in anxiety between pre- and post-intervention (F(1,16) = 6.79, *p* < 0.01; *t* = 6.14, *p* < 0.001; *t* = 4.16, *p* < 0.001—respectively, Tacón et al. [33], Sullivan et al. [32], Parswani et al. [27]). Furthermore, Keyworth et al. [35] reported a reduction in thoughts linked with worries (*t*(37) = 4.39, *p* < 0.001, *d* = 0.42). A significant difference between the groups in terms of patients’ perceived stress was also found in other studies (F = 107.62, *p* < 0.001; *t*(1,25) = 2.47, *p* < 0.001; *t*(1,102) = 3.10, *p* = 0.002—respectively, Momeni et al. [28], Nehra et al. [13], and Nyklíček et al. [31]).

Quality of life was reported by four studies. The summary effect size was very large, but in the random-effects model, it was not significant due to excessive variability among studies (*g* = 1.78, 95% CI: 0.06; 3.50, *p* = 0.043, *I^2^* = 98.1%, τ^2^ = 3.0) [10,17,30,31] (Table 4, Figure 2).

Other studies reported significant improvements in a variety of psychological factors, such as emotional control, coping strategies, and self-efficacy. For instance, Tacón et al. [33] revealed that participants of their mindfulness group showed an improvement in expressing negative feelings compared to pre-intervention (F(1,16) = 6.26, *p* < 0.02). Freedenberg et al. [21] have found that some styles of coping strategies (i.e., cognitive restructuring, acceptance, positive thinking, and distraction) increased significantly from before to after mindfulness interventions (F(1,44) = 5.191, *p* = 0.028). Specifically, significant differences and changes in direction were found for the reactive style (F(1,16) = 5.52, *p* = 0.03). In relation to perceived stress, the authors [21] found a significant decrease in stress after both the MBSR and online interventions (F(1,43) = 13.94, *p* = 0.001). As concerns self-efficacy, research conducted by Jalali et al. [30] revealed no significant differences between experimental and control groups, whereas significant changes in self-efficacy and quality of life occurred after 3 months, at the follow-up assessment (*p* < 0.01) [30].

As for the use of the online version of mindfulness training, some studies showed a significant effect specifically on stress (F(1,43) = 13.94, *p* = 0.001) and coping (F(1,44) = 5.91, *p* = 0.028) [21], mental functioning (*d* = 0.22, 95% CI 0.05; 0.38; *p* = 0.108) and depressive symptomatology (*d* = 0.18, 95% CI 0.02; 0.35; *p* = 0.143) [22].

### 3.5. Risk of Bias

For some studies, the methods of randomization were unclear. Furthermore, few studies described group allocation and randomization. Some RCTs reported adequate random sequence generation and allocation concealment and adequate blinding of participants’ personnel and outcome assessment [26]. Only a few studies had a low risk of bias in all other criteria [13,23].

## 4. Discussion

All studies included in this systematic review described relevant effects of MBI intervention on physical and/or psychological outcomes of patients with CVD. For physical outcomes, what clearly emerged from the meta-analysis is that MBI intervention has a real benefit in reducing systolic blood pressure, and the literature review suggests that heart palpitations and rate could also decrease. MBI had a moderate effect size in the reduction of diastolic blood pressure, but the meta-analysis was not significant. Moreover, for some physical risk factors, such as the impact of fatigue, unsteadiness/dizziness, breathlessness, and/or tiredness, an additional significant impact was also detected by a study [29]. As concerns psychological outcomes (such as depression, anxiety, stress, quality of life, styles of coping), all studies confirmed that different MBI-related approaches positively impact overall psychological wellbeing. Some studies found significant statistical effects for combined psychological conditions [9,13,27,28,30,31,32,33], whereas others found only a specific effect in one single domain, such as depression [10,22]. In contrast, Younge and colleagues [17] are the only ones who did not report significant differences before and after MBI in CVD patients. Nevertheless, the meta-analyses showed a moderate, large, and very large effect size for anxiety, perceived stress, and depression, respectively. As for quality of life, the random-effects model effect size was very large, but there were only four studies with a large variability; therefore, it was not significant.

Some researchers have highlighted how physiological measures (heart rate, blood pressure) can be influenced by mindfulness practice, suggesting significant improvements in pre- and post-treatment scores [17,22,25,26,27,29]. Furthermore, several studies have shown a direct correlation between mindfulness practice and scores related to psychological measures such as anxiety and depression [9,10,22,27,31,32]. The psychological wellbeing linked to mindfulness practice was also quantified in two studies that compared the scores relating to self-efficacy and perceived quality of life [29,30,32]. They showed an improvement in these aspects, thus highlighting the positive effects related to mindfulness on psychological wellbeing. However, Steffen et al. [24] and Younge et al. [17] did not report improvements in scores related to psychological measures, although better physiological outcomes (heart rate beat and blood pressure) were detected.

### Limitations

Part of the discrepancy reported in this systematic re-examination of literature may be dependent on some factors, such as (a) the different duration of MBI intervention (from 1 day to 12 weeks) [17,22], (b) the psychological battery, (c) the high dropout rates [10,17,26], and (d) the administration system. One of the main aspects that could influence the outcome of CVD patients is the duration of MBI treatments. Indeed, a wide heterogeneity characterized the literature where MBI-related training may last from 2 to 12 weeks, without any specific reference. Cullen et al. [36] claimed that to obtain significant statistical results, the training should last at least 8 sessions. Hence, when a shorter protocol has been used, the results are unreliable statistically [23]. Additionally, some studies evaluated mindfulness skills with specific surveys and scales [10], while others did not use any objective evaluation [23,25,26,29]. Moreover, in some studies, the training was led by a certified trainer [10,21,26,28,31], while in others, it was not [9,13,17,22,23,24,25,27,29,30,32,33]. Therefore, this aspect may have affected the impact of training on physical and psychological outcomes.

Another important aspect is the administration modality of MBI (in vivo vs. online). As regards online administration, there are few studies in the literature that have investigated this methodology [17,21,22], and the results of these studies are controversial because some studies showed a small positive effect on psychological outcomes (stress, mental functioning, and depressive symptomatology) [21], whereas others did not [17,22]. As regards physical outcomes, the same studies demonstrated a significant improvement in exercise capacity, systolic blood pressure, and heart rate [17,22]. These controversial results may depend on several factors: the age of the participants (adults vs. teenagers) and the methodology used (videoconference vs. online program). In addition, another facet of online training must be considered: the lack of external control. Indeed, teachers cannot control whether participants have practiced the tasks accurately. This can lead to less motivation and lower adherence than training with teachers and other group members [17]. Considering the paucity of studies on this approach, it is still not possible to recommend it. Finally, even if almost all of the studies reported the gender of participants, gender was not actually used as a factor for outcome analysis. Therefore, future studies should analyze gender differences for this kind of intervention.

## 5. Conclusions

This review was planned and conducted following best practice guidelines for systematic reviews and meta-analyses. Many RCTs in the general population as well as in high-risk disease groups were included. Subgroup analyses were conducted to assess the effects among these different participant groups. The applicability of the results was assessed. From the examined studies, convergent results are found. Despite some methodological differences (i.e., a few studies described group allocation, adequate random sequence generation, the predominant presence of women), we can affirm that there is sufficient evidence to conclude that MBI can be included in clinical recommendations for managing CVD symptoms. In future studies, it would be better to evaluate the effectiveness of online MBI applications, mainly in the context of a pandemic such as COVID-19, where there is a need to avoid direct contact between clinicians and patients and to reduce the number of admissions in hospital. Indeed, a middle way would therefore be ideal: an easily accessible online training but with more content and feedback from a trainer. It will also be important to replicate physiological and psychological outcome measures not included in the meta-analyses while increasing the diversity of the setting, methods, and population to include them as moderators. Again, other important aspects remain to be established, such as the use of measures on mindfulness skills, teacher competency, and the adherence to the intervention protocol to disentangle therapist effects from intervention–content effects.

## Figures and Tables

**Figure 1 brainsci-11-00727-f001:**
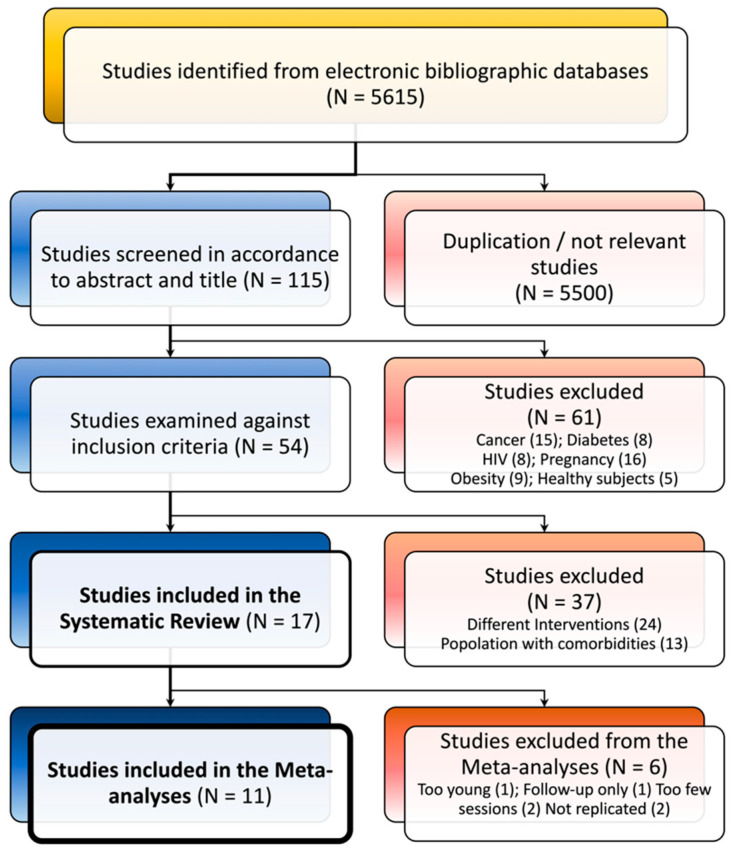
Flowchart of studies’ identification.

**Figure 2 brainsci-11-00727-f002:**
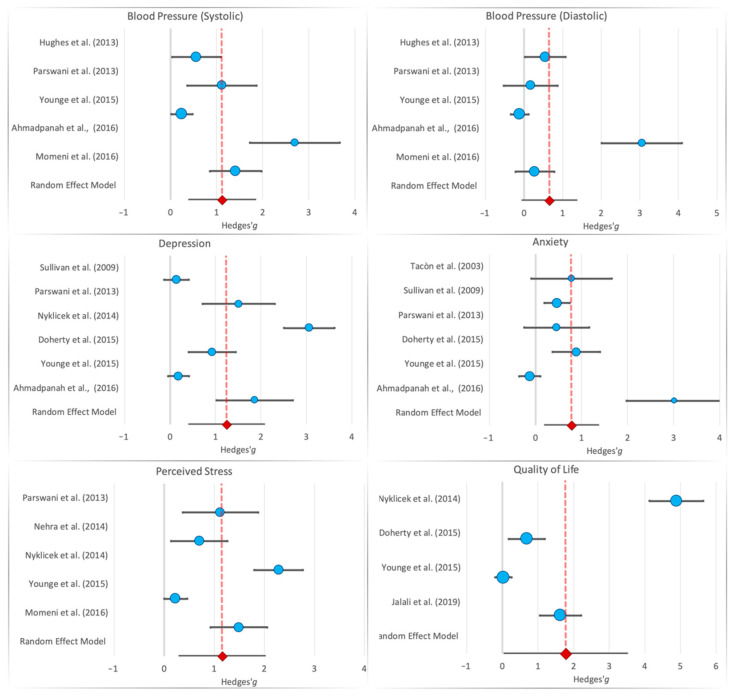
Forest plot for the six selected constructs. Dotted line is the Hedges’ *g* for the random-effects model [9,10,17,26,27,28,30,31,32,33].

**Table 1 brainsci-11-00727-t001:** Keywords and example of Search strategy.

MeSH Terms	PubMed	Science Direct	Scholar
“Heart Diseases” AND “mindfulness”	11/15	4/1187	7/35,900
“Cardiovascular Diseases” AND “Mindfulness”	23/33	0/1934	23/34,900
“Heart Failure” AND “Mindfulness”	1/2	9/2168	1/36,500
“Cardiovascular failure and Mindfulness”	-----	6/1123	5/24,200
“Congestive heart failure and Mindfulness”	3	3/444	2/6510
“Coronary Disease” AND “Mindfulness”	3/3	0/79	11/20,500
“Cardiovascular Abnormalities” AND “Mindfulness”; Congenital Abnormalities” AND “Mindfulness”; “Heart Defects, Congenital” AND “Mindfulness;” Cardiovascular Infections” AND “Mindfulness”	------	3/621	------

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
