# Peer review of "Mindfulness-Based Interventions for Physical and Psychological Wellbeing in Cardiovascular Diseases: A Systematic Review and Meta-Analysis"

_brainsci, 2021, doi:10.3390/brainsci11060727_

Round 1

Reviewer 1 Report

The paper is very well written and could become a valuable reference in this field of research. The authors have adopted high-quality standards and describe very well the rationale and procedure used in their study.

In terms of presentation I found the figures (1 & 2) to be very difficult to read due to low quality. This must be improved in the actual publication.

I was a bit surprised to see that conference papers were excluded. In my opinion there are high quality conferences that live up to high review standards.

The sentence on line 114 "studies should be based on intervention protocol" needs to be better explained (it should also be "an intervention").  Does this imply that only studies with a control & experimental condition are included or that there should be an explicitly described protocol?

On line 166 it was not clear if the study that did not report gender information was excluded? Since the next sentence is "Of the remaining..." I assume it was.

The sentence on line 172 could be rephrased. It is now "One study was the only one...".

The statement on line 209 (that only 5 studies found a certain effect) made me wonder if it would not be valuable to compare those 5 studies with others that did not find this effect? Similar for the study reported on line 288. Assuming that all studies are correct, the differences (e.g. intervention type or sample characteristics) could yield valuable insights for better understanding such effects.

On line 214 a statistical result for reference 31 is provided. Consulting the original reference I found a different effect:
The groups were not different on almost all demographic, medical, psychological well-being, mindfulness, and quality of life variables (p[.10), except for indication for PCI (which was an acute condition in 23 participants in the group condition vs. 9 in the self-help condition; v2 (1) = 6.54, p = .01), and perceived stress (being higher in the group condition; t (105) = 2.06, p = .04; see Table 2 for means and SD). A tendency was found for the group condition to score higher on symptoms of anxiety and depression at baseline com- pared to the self-help condition (t (105) = 1.84, p = .07).

Can the authors explain this difference?

The discussion and conclusion are insightful and the authors have well articulated the limitations of the study. 

One aspect I missed was any discussion on gender differences. It is known that such differences are present and can influence study outcomes for this topic. Do the selected papers provide any basis for elaborating on this?

Author Response

Dear Reviewer

We thank you for comments and suggestions.

We improved the quality of the figures 1 and 2. Figure 1 was replaced with a better one.

As regards conference papers, we agree with you, but actually we wanted to be very tight with the criteria, which is also reflected in the rest of the paper.

As regards the intervention protocol, we included only studies reporting an explicit intervention protocol tested against a control group. The text was updated accordingly.

As regards gender information, 16 out of 17 studies reported the gender of participants (women vs men). The text was updated accordingly.

The sentence "One study was the only one..." was rephrased.

As regards the statements on lines 209 and 288 we have reported the number of studies where an effect was found in the meta-analysis for each single variable; for the other studies which did not test the same variable, a direct comparison of that variable is not possible to be reported. We have clarified the text accordingly.

As regards the text on line 214, we thank you for noticing an error. A few numbers were incorrectly reported. We have corrected the text accordingly.

As regards gender differences, at the end of “4.1 Limitations” we added “Finally, even if almost all of the studies reported the gender of participants, actually gender has not been used as a factor for outcome analysis. Therefore, future studies should analyze gender differences for this kind of intervention.”

English was checked by an English editing service.

Best regards

Reviewer 2 Report

I found your review of the literature regarding MBI and cardiovascular disease to be well written and of sound methodology. My only recommendation is to have an editor review your manuscript for English grammar and style.

Author Response

Dear Reviewer

we thank you for your report, we really appreciate.

English was checked by an English editing service.   Best regards